# Use of an Online Platform to Evaluate the Impact of Social Distancing Measures on Psycho-Physical Well-Being in the COVID-19 Era

**DOI:** 10.3390/ijerph19116805

**Published:** 2022-06-02

**Authors:** Stefania Scuri, Marina Tesauro, Fabio Petrelli, Ninfa Argento, Genny Damasco, Giovanni Cangelosi, Cuc Thi Thu Nguyen, Demetris Savva, Iolanda Grappasonni

**Affiliations:** 1School of Medicinal and Health Products Sciences, University of Camerino, 62032 Macerata, Italy; stefania.scuri@unicam.it (S.S.); ninfa.argento@studenti.unicam.it (N.A.); genny.damasco@studenti.unicam.it (G.D.); iolanda.grappasonni@unicam.it (I.G.); 2Department of Biomedical, Surgical and Dental Sciences, University of Milan, 20122 Milan, Italy; marina.tesauro@unimi.it; 3Units of Diabetology, ASUR Marche, Area Vasta 4, 63900 Fermo, Italy; giovanni.cangelosi@virgilio.it; 4Department of Pharmaceutical Administration and Economics, Hanoi University of Pharmacy, Hanoi 10000, Vietnam; cucnguyen.pharm@gmail.com; 5Plastic Reconstructive and Aesthetic Surgery, Nicosia General Hospital, Nicosia 2029, Cyprus; drsavvademetris@gmail.com

**Keywords:** SARS-CoV-2, public health, epidemiology, quality of life, mental health, PTSD

## Abstract

*Background:* The SARS-CoV-2 pandemic (Severe Acute Respiratory Syndrome Coronavirus 2) and the worldwide health crisis have significantly changed both people’s habits and lifestyles. Most of the studies found in the literature were carried out on specific professional categories in the socio-health sector, taking into consideration psychological disorders in relation to work. The purpose of this study was to analyze the psychological impact on a portion of the normal population subjected to lockdown. *Methods:* A questionnaire was distributed in the period between 23 March 2020 and 18 May 2020 (during Italian lockdown) using an online platform. The scales GAD-7, IES-r, PHQ-9 and MANSA were used to investigate the level of anxiety, the presence of post-traumatic stress disorder, the severity of depression and the perceived quality of life, respectively. *Results:* Four hundred and eight Italian subjects responded. Females and younger people were more affected by anxiety and depression. Post-traumatic stress disorder affected about 40% of the population sample, significantly young people and women, thus attesting to an important psychopathological response. About one-fifth of the sample population recorded an unsatisfactory quality of life. *Conclusions:* The results highlight the need to set up preventive interventions (primary and secondary), trying to focus on the most fragile group of subjects from a psychosocial point of view, in order to obtain a significant reduction in psychophysical damage in terms of relapses and outcomes.

## 1. Introduction

In March 2020, WHO declared the disease caused by the new coronavirus SARS-CoV-2 [1] as a global pandemic, and several governments implemented national quarantines to control the spread of the infection. The Italian Government imposed a national lockdown on 9 March 2020 [2,3,4].

These measures included the maintenance of social distancing, the avoidance of gatherings, working and studying from home, blocking all unnecessary economic activities (with severe economic and job crises) and a general reduction in free movement. Restrictions on travel, social contacts and emotional relationships, the reduction in or loss of jobs and domestic confinement, fear of contracting the infection, the absence of vaccines and effective medical treatments in the first period, uncertainty about the future, etc., have contributed to accentuating the sense of loneliness, abandonment and anxiety in the general population [5,6,7]. While confinement and isolation have proven highly effective in limiting the transmission of the virus, they could have negatively impacted the mental health of the global population, with a worrying increase in psychiatric symptoms [8]. Beyond the intrinsic stress of the disease itself, depression, irritability, insomnia, fear, confusion, anger, frustration, boredom, memory disorders and confusion have emerged [9]. The SARS-CoV-2 pandemic (COVID-19) and the health emergency have significantly changed both people’s habits and lifestyles. The choices made by governments have produced new living and working conditions that, combined with the obvious fear of contagion and disease, may have induced stressful situations with possible psychological distress in the general population [10,11,12,13,14,15,16]. Whatever the nature of the traumatic event, this can lead to the development of a series of disorders, including post-traumatic stress disorder (PTSD) [17,18,19,20]. PTSD is a form of mental distress that can occur in people of all ages and can develop as a result of highly traumatic experiences, as well as from repeated and continuous exposure to episodes of violence and decay. The literature highlights a negative impact on the mental health of those who underwent unexpected events, such as earthquakes, floods, terrorism, etc. In the last years, studies have underlined that some epidemic situations have a strong impact on the quality of life and the psychological well-being of individuals, including anxiety, mood disorders, psychological distress, sleep disturbances and post-traumatic symptomatology, proving that pandemics can also lead to PTSD [21,22,23,24,25,26,27].

Most of the studies were carried out on specific professional categories in the socio-health sector, taking into consideration psychological disorders in relation to work [28,29,30,31,32,33,34,35]. Several studies have investigated the aspects of the psychophysical well-being of the general population as a consequence of restrictive measures to counteract the spread of coronavirus [5,6,8,9,36,37,38,39,40].

The purpose of this study was to analyze the psychological situation in a part of the general Italian population during the restrictive measures of the lockdown period. This study aimed to add information relating to the psychological situation of the population of Italy—one of the most affected countries by the pandemic—in a period in which sufficient information about the transmission, prevention and treatment of coronavirus disease was not yet readily available.

## 2. Materials and Methods

This paper analyzes socio-psychological issues related to the COVID-19 emergency. The analysis was conducted on a population sample during the lockdown period, from 23 March to 18 May 2020, using an online prepared interview in the form of a questionnaire. Information about the possibility of filling in the online questionnaire was advertised by messaging apps (i.e., WhatsApp) and social networks (i.e., Facebook), asking people to spread the information to their contacts.

The choice to use an online platform to administer the questionnaire was due to the difficulty of obtaining information through direct interviews, given the impossibility of free movement during the lockdown period [41,42].

Unlike a paper questionnaire, which may happen to have variable or partial answers, the use of an online questionnaire has many advantages, including (a) more people can be reached remotely, (b) the full completion of all questions is obtained (in the online questionnaire, the failure to fill in one or more items is reported to the responder, and he/she cannot proceed with the rest of the questions), (c) the responder “feels safe” in the sense that he/she does not feel judged by the interviewer, nor does he/she feel that his/her privacy is violated, and (d) the online system automatically records and transfers all of the answers to an Excel file, without any manual data entry errors that can arise when performed by an operator.

The “Google forms” platform was therefore used, and the questionnaire consisted of a section to collect personal information, a second part investigating changes in habits during the lockdown and the final part containing the selected rating scales, investigating the possible repercussions of the lockdown on the mental health of the individuals.

The psychiatric assessment scales were GAD-7 (General Anxiety Disorder Assessment) [43,44], which investigates the level of anxiety and concern felt by the patient, IES-R (Impact of Event Scale) [45,46,47], which evaluates post-traumatic stress disorder, PHQ-9 (Patient Health Questionnaire) [48,49], which is used for the determination of the severity of depression, and the MANSA (MANchester Short Assessment of quality of life) scale [50], which is used to assess the perceived quality of life.

The GAD includes 7 items on a 4-point Likert scale ranging from 0 (never) to 3 (nearly every day). The total score ranges from 0 to 21, with higher scores indicating more severe functional impairments as a result of anxiety. For this study, we defined a GAD-7 total score of 10 points or higher as an index of the presence of generalized anxiety symptoms. IES-R is a scale developed to assess symptoms of PTSD. It is a 22-item scale measuring three core phenomena of PTSD: re-experiencing traumatic events, defensive avoidance and denial of trauma-related memories and emotions, and hyperarousal. Respondents are asked to rate each item in the IES-R on a scale of 0 (not at all), 1 (a little bit), 2 (moderately), 3 (quite a bit) and 4 (extremely) in relation to the past 7 days. In this study, the total score of the scale was used, and a cut-off point of 1.5 (equivalent to a total score of 33) was employed. The PHQ-9 scale consists of 9 items that correspond to the symptoms of major depression according to the DSM-IV (Diagnostic and Statistical Manual for Mental Disorders, volume 4). The score range is between 0 and 27. Scores between 0 and 9 indicate the presence of subthreshold depression. A score of 10 is indicated as the point where the sensitivity and specificity of the test highlight depression of clinical relevance. MANSA is defined as the mean score of 12 satisfaction ratings in different life domains and life in general. Each item is rated on a Likert-type scale ranging from 1 (lowest satisfaction) to 7 (highest satisfaction) with 4 as a neutral middle point. 

The questionnaire validity was tested by administering it to 15 people of different socioeconomical and cultural statuses to evaluate the “face validity” [51,52].

In the initial part of the online questionnaire, the aim of the survey was briefly explained, and electronic informed consent was requested from each participant before starting the completion of the questionnaire. 

The study was conducted according to the Helsinki Declaration. This study was approved by the ethics committee of the Marche Region (CERM-2018 64 F99-F99), Italy.

The data were processed using an Excel Workbook (Microsoft Office, Redmond, VA, USA) and analyzed using XLSTAT Software (Addinsoft Inc., New York, NY, USA and Origin Pro 9.1 (Origin Lab, Northampton, MA, USA) [53,54].

Continuous variables are summarized using mean and standard deviation (SD), and categorical variables are reported using frequencies and percentages. The association between gender and age and levels of anxiety, depression, stress and quality of life were analyzed using the chi-square test, considering *p* < 0.05 as a statistically significant result. 

## 3. Results

Four hundred and eight Italian subjects aged between 18 and 79 (37.54 ± 14.45) responded to the questionnaire. Table 1 shows the demographic characteristics of the interviewed people.

Table 2 shows the answers of interviewed people when asked, “since when do government restrictions come into effect, how often do you do the following activities”, and highlights that several of the listed activities increased (reading a book, cooking, lounging, difficulty sleeping and smoking), were reduced/interrupted (physical activity and sleep) or remained unchanged (smoking).

The answers regarding changes in daily activities during the lockdown were analyzed to highlight possible differences between gender and age groups. 

Among the sample, 182 subjects (44.61%) reduced/stopped physical activity (16.67% male and 27.94% female; NS) (40.54% 18–30 years, 48.87% 31–50 years, 46.67% >50 years; NS), 66 (16.18%) increased smoking (5.39% male and 10.78% female; *p* < 0.05) (21.08% 18–30 years, 15.03% 31–50 years, 7.78% >50 years; *p* < 0.05), and 104 (25.49%) had sleep disturbances (9.31% male and 16.18% female; NS) (22.16% 18–30 years, 25.56% 31–50 years, 32.22% >50 years; NS). 

The analysis of the PHQ9 scale, used to highlight situations of depression, showed that 149 subjects (36.52% of the sample) exceeded the cut-off of 10. Of these, 78 (52.35%) were female and 71 (47.65%) were male.

From the GAD-7 scale, which highlights states of anxiety, it emerged that 170 subjects (41.67%) had a minimum level of anxiety (score 0–4), 136 had a medium level (33.33%) (score 5–9), 65 had a moderate level (15.93%) (score 10–14) and 37 (9.07%) had severe anxiety (≥15).

The IES-r scale, which evaluates PTSD, showed that 154 subjects (37.75%) exceeded the cut-off of 33 (115 females and 39 males).

The MANSA scale, which assesses the quality of life, showed that 85 subjects (59 women and 26 men) (20.83%) had a score of <4, indicating a low quality of life.

In Table 3, in relation to the evaluation scales used, it is highlighted that many of those who had reduced physical activity, increased use of cigarettes and reduced time dedicated to rest exceeded the cut-off of the evaluation scales used.

An evaluation of the IES-R scale was also carried out by calculating the average score, whose cut-off is represented by the value 1.5. The total sample obtained an average score of 1.20 ± 0.89, while among those who passed the cut-off of 33 as a total score (of 1.5 as the average score), the average score was 2.16 ± 0. 49.

Regarding the IES-R scale, the sample was further analyzed considering the following three subscales:

Avoidance (AV), i.e., insensitivity and detachment (for example, avoiding thoughts, emotions, people or situations related to the event).

Intrusion (Intrusion, IS), i.e., the presence of sudden and recurring flashbacks, intrusive memories and dreams whose content and aroused emotions are attributable to the traumatic experience. In addition, a marked and amplified stress response to internal or environmental stimuli that recall or are associated with the trauma can be observed.

Hyperarousal (Hyperarousal, HYP), i.e., sleep disturbances (both falling asleep and maintaining sleep), irritable behavior and sudden outbursts of anger towards others even in the absence of provocation, increased sensitivity to potential threats, and particular reactivity to facing unexpected stimuli.

Using the subscales indicated above, the sample exceeding the cut-off was analyzed. As can be seen from the results, differences are evident, showing values well above 1.5 in the group that exceeded the cut-off, both in males and females (Table 4).

It can be observed that in subjects who passed the cut-off, all three subscales showed average values well above 1.5, considered a demarcation point between normal and risk situations.

Comparing the interviewed sample and taking into account gender and age groups, Table 5 reports the results obtained by the subjects who passed the cut-off of the various evaluation scales. It is observed, for example, that for all scales, with the exception of MANSA, the youngest category was the most compromised (*p* < 0.05). Considering gender, males were more prone to anxiety problems, while females were more prone to depression and PTSD (*p* < 0.05). No significant differences were highlighted for the MANSA scale, with which very similar results were obtained between the groups considered.

## 4. Discussion

Our study on the impact of the COVID-19 emergency on the quality of life and mental health of the population shows that relational closure involves negative reactions that lead to psychopathological distress.

In the sample that we evaluated during the lockdown, the quality of life appears to be compromised, mainly in women and young people, the categories most affected by post-traumatic stress disorder (PTSD), which produces a significant negative impact on the quality of life and on the social and occupational functioning of individuals. Normally, the home environment determines situations of relaxation and distraction. On the contrary, difficulty sleeping and the need to smoke, as affirmed by the interviewed people, signal discomfort that also affects a family environment. The increase in cooking activity during the lockdown can have an ambiguous meaning. On the one hand, it could be an indication of relaxation and attention to a healthier diet; on the other, if correlated with reduced physical activity and increased laziness, it could contribute to weight gain and consequently to health risks such as cardiovascular diseases, metabolic risk, cancer, etc. [55,56,57,58]. 

Many studies have been conducted since the declaration of the pandemic. These studies aimed to investigate the effects of the pandemic, especially in healthcare workers. Restrictions and quarantine were found to have a major impact on mental health with major psychological implications, such as stress, anxiety, depression and feelings of frustration [29,30].

The increase in new cases has induced generalized fear and feelings of anxiety, and reactions have been seen to range from panic (or collective hysteria) to pervasive feelings of hopelessness associated with negative outcomes, including suicidal behavior [15,16,59]. Some researchers have turned their attention to the impact of the quarantine, trying to identify actions to contain the mental health problems resulting from it [27]. At the same time, information and education campaigns have been adopted in various countries, which have produced containment effects on these problems, since they have succeeded in making the population question and abolish these new habits [60,61].

Correct information and targeted communications have been found to be useful in increasing the knowledge of actions to be taken in case of positivity in symptomatic subjects or when people have been in contact with a possible COVID-19 case.

Four hundred and eight individuals responded to the questionnaire, making it possible to carry out an initial assessment of the effects that the COVID-19 emergency can have on the quality of life and mental health of the generalized population.

Analyzing the results and the responses to the rating scales, there was a significant increase in smoking among females (*p* < 0.05) and in the 18–30 age group (*p* < 0.05). In addition, women and young people were more affected by anxiety and depression. Post-traumatic stress disorder affected about 40% of the sample, significantly young people and women, thus attesting to an important psychopathological response. About one-fifth of the interviewees recorded an unsatisfactory quality of life, with no significant differences between males and females or among different age groups.

The results obtained, which show greater problems for young people and women, could be explained by the fact that these categories have suffered more from the changes imposed by the restrictive measures. The effects on women are probably also due to the greater load of commitment linked to the continuous presence of all family members in the home, with greater tasks also in terms of distance learning, remote working and a complete change in the organization of the day; the effects on young people are likely due to reduced sociability, the inability to meet with their peers, distance learning and reduced sporting activity.

The increase in psychological distress seems to be related to the reduction in the enjoyment of normal daily activities, as well as to greater difficulties in concentrating, lack of rest, etc.

Some limitations of our study can include the way that the questionnaire was advertised and the age groups of the subjects who completed our survey. More specifically, the fact that this study was advertised through social platforms and messaging applications, as well as word-of-mouth, puts a limit on the older population that does not have access or does not know how to use these applications or fill in an online questionnaire in general. Hence, our age groups are more concentrated in the younger population under 50 years of age. Since almost half of our subjects are younger than 30 years of age, this can also place some limitations on the results of our study. On the other hand, this is totally understandable since younger people have more access to social platforms and more free time, and it is much easier and faster for them to complete an online questionnaire compared to older generations. 

## 5. Conclusions

The COVID-19 pandemic represents an experience with traumatic implications involving various groups of the population, which puts each citizen in a vulnerable position and potentially at risk of suffering pathological reactions, even if people have not experienced or suffered exposure to the virus.

The variety of traumas, which cause emotional fluctuations, anxiety, concern for one’s family and the economic crisis, a decrease in social contacts and the loss of interest in common daily activities, is not a negligible element.

The analysis carried out shows an increase in psychological distress and a decrease in the perception of the quality of life. Many of those who have been exposed to this stress risk psychopathological outcomes, which, if underestimated, can proceed to a decrease in relationships and psychosocial functioning and, consequently, an increased possibility of developing real disorders with pathological behaviors of various kinds.

Clear, effective and rapid communication can be a valuable aid to people in promoting adherence to correct behaviors, which would decrease the sense of anguish, also reducing long-term complications. Correct information therefore represents the key to increasing the level of resilience.

It would be appropriate to integrate healthcare with medical-psychiatric care for all, not only for those who work in the socio-health fields. Expanding knowledge on the impact of the disease on mental/neurological health is also of fundamental importance.

What emerged highlights the need to set up more preventive interventions (primary and secondary), trying to work on the most fragile subjects from a psychosocial point of view, in order to obtain a significant reduction in psychophysical damage in terms of relapses and outcomes.

It is important to identify the subgroups of the population most exposed to negative behavior during health emergencies in order to correctly inform policy makers to plan effective public health actions. Finally, analyzing lifestyle changes during an emergency can be a necessary tool so as to improve the organization and public health interventions.

## Figures and Tables

**Table 1 ijerph-19-06805-t001:** Characteristics of the sample.

	n.	%
**Gender**		
Female	252	61.76
Male	156	38.24
**Age**		
18–30	184	45.10
31–50	133	32.60
>51	90	22.30
**Employment status**		
Employed	256	62.75
Unemployed	37	9.07
Student	91	22.30
n.d.	24	5.88
**Marital status**		
Single	108	26.47
Engaged	140	34.31
Married	146	35.78
Divorced	14	3.43
**Educational level**		
Elementary	4	0.98
Middle school	51	12.50
High school	179	43.87
Graduation	57	13.97
Master’s degree	77	18.87
Postgraduate	40	9.80

**Table 2 ijerph-19-06805-t002:** Changes in daily activities during the lockdown.

	Increased N (%)	Reduced N (%)	Interrupted N (%)	Never Played N (%)	Unchanged N (%)
Read a book	147 (36.03)	47 (11.52)	9 (2.21)	111 (27.21)	94 (23.04)
Cooking	218 (53.43)	22 (5.39)	6 (1.47)	94 (23.04)	68 (16.67)
Physical activity	79 (19.36)	142 (34.80)	40 (9.80)	82 (20.10)	65 (15.93)
Smoking	66 (16.18)	27 (6.62)	3 (0.74)	170 (41.67)	142 (43.80)
Laze about	136 (33.33)	38 (9.31)	9 (2.21)	154 (37.75)	71 (17.40)
Sleeping	120 (29.41)	92 (22.55)	12 (2.94)	99 (24.26)	85 (20.83)

**Table 3 ijerph-19-06805-t003:** Changes in daily activities during the lockdown in people who exceed the cut-off of the evaluation scales.

	PHQ-9 ≥ 10 N (%)	M N (%)	F N (%)	GAD-7 ≥ 10 N (%)	M N (%)	F N (%)	IES-R ≥ 33 N (%)	M N (%)	F N (%)	MANSA < 4 N (%)	M N (%)	F N (%)
Reduced/interrupted physical activity	73 (17.89)	36 (23.08)	37 (14.68)	44 (10.78)	11 (7.05)	33 (13.1)	75 (18.38)	21 (13.46)	54 (21.43)	43 (10.54)	10 (6.41)	33 (13.1)
Increased smoking	39 (9.56)	14 (8.97)	25 (9.92)	32 (7.84)	10 (6.41)	22 (8.37)	38 (9.31)	10 (6.41)	28 (11.11)	22 (5.39)	6 (3.85)	16 (6.35)
Reduced/interrupted sleep	37 (9.07)	14 (8.97)	23 (9.13)	31 (7.59)	7 (4.49)	24 (9.52)	44 (10.78)	12 (7.69)	32 (12.7)	31 (7.60)	10 (6.41)	21 (8.33)

**Table 4 ijerph-19-06805-t004:** Values obtained in the three IES-R subscales, both in the total sample and in the group that exceeded the cut-off of 33 (or the average value of 1.5).

IES-R Subscales	Average Score ± SD	Male Average Score ± SD	Female Average Score ± SD
Total sample			
AVOIDANCE	1.20 (0.92)	0.90 (0.83)	1.38 (0.93)
INTRUSION	1.17 (0.92)	0.92 (0.81)	1.33 (0.95)
HYPERAROUSAL	1.23 (0.98)	0.90 (0.87)	1.43 (0.99)
Sample with cut-off ≥ 33			
AVOIDANCE	2.10 (0.62)	1.99 (0.60)	2.14 (0.63)
INTRUSION	2.16 (0.56)	2.08 (0.44)	2.18 (0.60)
HYPERAROUSAL	2.23 (0.64)	2.09 (0.58)	2.28 (0.65)

**Table 5 ijerph-19-06805-t005:** Analysis of the sample that exceeded the cut-offs of the 4 evaluation scales. Differences grouped according to age group and gender.

	PHQ-9 ≥ 10 Average Score ± SD	GAD-7 ≥ 10 Average Score ± SD	IES-R ≥ 1.5 Average Score ± SD	MANSA < 4 Average Score ± SD
AGE				
18–30	15.38 ± 4.52	14.00 ± 3.03	2.19 ± 0.50	3.01 ± 0.76
31–50	14.87 ± 4.68	14.05 ± 3.91	2.13 ± 0.47	2.74 ± 0.99
>50	14.40 ± 3.81	12.4 ± 2.63	2.08 ± 0.50	2.43 ± 1.02
GENDER				
Male	15.79 ± 5.00	13.13 ± 2.86	2.05 ± 0.41	2.84 ± 0.91
Female	14.46 ± 3.89	14.08 ± 3.27	2.19 ± 0.52	2.82 ± 0.90

## Data Availability

The data presented in this study are available on request from the corresponding author.

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
