# Peer review of "Use of an Online Platform to Evaluate the Impact of Social Distancing Measures on Psycho-Physical Well-Being in the COVID-19 Era"

_ijerph, 2022, doi:10.3390/ijerph19116805_

Round 1

Reviewer 1 Report

1. The author(s) might need to strengthen the logical flow and contents of the introduction. in particular, the author(s) must write a bit more about why this study is needed and the potential consequences of COVID lockdown.

2. please include either research questions or hypotheses, preferably hypotheses and the process of hypotheses development.

3. for the data collection, please write the response rate and sampling strategy

4. For table 2, please do the statistical analysis and see whether it is really changed.

5. In Table 4, the author(s) reports the total sample and 'sample with cut-off >=33'. why does it need and how to interpret it?

Author Response

  1. The author(s) might need to strengthen the logical flow and contents of the introduction. in particular, the author(s) must write a bit more about why this study is needed and the potential consequences of COVID lockdown.

We have explained the italian lockdown, and have reported information about the impact on the quality of life and the psychological well-being following disasters and epidemic situations (see pag. 2)

  1. please include either research questions or hypotheses, preferably hypotheses and the process of hypotheses development.

We have added information about this request (see pag. 2)

  1. for the data collection, please write the response rate and sampling strategy

We cannot indicate the response rate because people was free to fill-in the online questionnaire. 

  1. For table 2, please do the statistical analysis and see whether it is really changed.

We have done the statistical analysis to evaluate differences between age groups and gender, and have reported the results on page 5

  1. In Table 4, the author(s) reports the total sample and 'sample with cut-off >=33'. why does it need and how to interpret it?

We have explained why we wrote Table 4, adding the following sentence:

“It can be observed that in subjects who pass the cut-off, all three subscales show average values well above 1.5, considered as a demarcation point between normal and risk situations”.

If you think this is not necessary, we can delete it

Reviewer 2 Report

First, I would like to thank the opportunity of reading this work. I found the study interesting, despite there being already some studies on mental health during the coronavirus lockdown. I think the authors could highlight more, since the beginning of the paper,  the novelty of the study and the contribution to the literature or the potential practical implications of this study. In general, I found the literature review can be deepened. From the reader's perspective, we need to read the results section to be clearer about what mental health outcomes are being analyzed in this study. The authors measured several mental and physical health indicators, and this is a clear contribution of this paper to the literature that must be highlighted. Also, they control whether there are differences depending on if participants are men or women and their age. What previous literature says about the contribution of demographic variables in the studied variables? This is something the authors could introduce in the literature review section.

Moreover, I have some specific comments hoping they contribute to the improvement of the paper.

  • There are some typos and spelling errors in the paper. Example – Line 20: “408 subjects risponded”. Please, amend.
  • Abstract: Instead of mentioning the scales used (GAD-7, IES-r, PHQ-9 and MANSA), I believe it will be more interesting, from the readers' perspective, to identify the variables analyzed through these scales. More precisely, the authors could add this information when they mention the study's purpose in the abstract section: “The purpose of this study is to analyze the psychological situation in a part of the normal population subjected to lockdown.” Also, the following sentence adds nothing new to what was previously mentioned, and I suggest removing it: “with the purpose of investigating the possible repercussions of the lockdown on the mental health of the interviewees”. In addition, the authors could add information about the participants' country of origin, and the period when the data were collected in the abstract (latter on, I see data were collected in march-may 2020).
  • Literature review could be deepened. For instance, the authors mentioned Post-Traumatic Stress Disorder (PTSD). I missed, at least, a definition of what it consists of. In addition, if available, it will be interesting to add some statistics regarding the mental health of the studied population before the coronavirus pandemic.
  • Materials and Methods: Line 59 – “prepared online interview”. The authors conducted remote interviews or applied an online questionnaire? Please, clarify.
  • Materials and Methods: Line 88 – “This study was approved by the ethics committee of the Marche Region (CERM-2018 64 F99-F99)”. Please, add also the country (Italy).
  • Results: Line 96 – “408 subjects”. Please, add the country where the subjects are from.
  • Results: It will be important to add some information about the psychometric properties of the scales used, a sample of items, and their adequacy to use in the “normal” population.
  • Discussion: The authors could recognize the limitations of the present study and the avenues for future studies, and practical implications.

Author Response

There are some typos and spelling errors in the paper. Example – Line 20: “408 subjects risponded”. Please, amend.

The English of the manuscript was edited from by a native English-speaking

Changes were done in all text.

Abstract: Instead of mentioning the scales used (GAD-7, IES-r, PHQ-9 and MANSA), I believe it will be more interesting, from the readers' perspective, to identify the variables analyzed through these scales. More precisely, the authors could add this information when they mention the study's purpose in the abstract section: “The purpose of this study is to analyze the psychological situation in a part of the normal population subjected to lockdown.” Also, the following sentence adds nothing new to what was previously mentioned, and I suggest removing it: “with the purpose of investigating the possible repercussions of the lockdown on the mental health of the interviewees”. In addition, the authors could add information about the participants' country of origin, and the period when the data were collected in the abstract (latter on, I see data were collected in march-may 2020).

changes were done (see pag. 1)

Literature review could be deepened. For instance, the authors mentioned Post-Traumatic Stress Disorder (PTSD). I missed, at least, a definition of what it consists of. In addition, if available, it will be interesting to add some statistics regarding the mental health of the studied population before the coronavirus pandemic.

We have added information about PTSD (see pag 2).

We do not have information about statistics regarding mental health before the pandemia

Materials and Methods: Line 59 – “prepared online interview”. The authors conducted remote interviews or applied an online questionnaire? Please, clarify.

We have clarified that we used an online questionnaire (see pag. 2, line 85-88)

Materials and Methods: Line 88 – “This study was approved by the ethics committee of the Marche Region (CERM-2018 64 F99-F99)”. Please, add also the country (Italy).

We have added the requested information (see pag. 3, line136)

Results: Line 96 – “408 subjects”. Please, add the country where the subjects are from.

We have added the requested information (see pag. 4, Line 144)

Results: It will be important to add some information about the psychometric properties of the scales used, a sample of items, and their adequacy to use in the “normal” population.

We have added the requested information (see pag. 3, line 111-128)

Discussion: The authors could recognize the limitations of the present study and the avenues for future studies, and practical implications.

We have added the requested information (see pag 8, line298-302)

Round 2

Reviewer 1 Report

I do appreciate the authors for their hard work in addressing all comments from reviewers.

Author Response

Dear Author , thank you very much for your time taken for reviewing our manuscript.

Reviewer 2 Report

I would like to congratulate the authors on this improved version of the paper. The authors globally addressed my previous comments and I have nothing more substantial to add. I just leave some “little things” I found to amend:

Line 94: The following sentence: “obtaining the full completion of all the questions (in the online questionnaire the failure to fill in one or more items is reported to the responder and he” I would suggest replacing “he” to “they” or add “he/she”. Also the same within the following sentences, for instance, replace: the responders "feel safe" in the sense that they do not feel judged by the interviewer nor do they feel their privacy violated…

Line 98: “without any manual data entry errors by an the operator”. Please an article: an or the?

Although the authors addressed avenues for future studies, and practical implications, I am still missing a brief description of the potential limitations of the present study. 

Author Response

Dear reviewer, thank you very much for your time and your modifications for our paper.

I would like to congratulate the authors on this improved version of the paper. The authors globally addressed my previous comments and I have nothing more substantial to add. I just leave some “little things” I found to amend:

The whole manuscript was reproofed, and minor English modifications were done.

Line 94: The following sentence: “obtaining the full completion of all the questions (in the online questionnaire the failure to fill in one or more items is reported to the responder and he” I would suggest replacing “he” to “they” or add “he/she”. Also the same within the following sentences, for instance, replace: the responders "feel safe" in the sense that they do not feel judged by the interviewer nor do they feel their privacy violated…

The whole paragraph is modified to include he/she adverbs instead of only he. Also, some minor grammatical modifications made to this paragraph, highlighted in blue colour, and strikethrough what is needed to be deleted.

Line 98: “without any manual data entry errors by an the operator”. Please an article: an or the?

 Correction to AN operator and minor English modification of the sentence.

Although the authors addressed avenues for future studies, and practical implications, I am still missing a brief description of the potential limitations of the present study. 

Limitations of the study have been added in the discussion section of our study.